# Multimodal Communication and Peer Interaction during Equation-Solving Sessions with and without Tangible Technologies

**Daranee Lehtonen** [1,*] , **Jorma Joutsenlahti** [2] **and Päivi Perkkilä** [3]

1    Faculty of Information Technology and Communication Sciences, Tampere University, Kanslerinrinne 1, P.O. Box 300, 33014 Tampere, Finland

2    Faculty of Education and Culture, Tampere University, Åkerlundinkatu 5, P.O. Box 700, 33014 Tampere, Finland

3    Kokkola University Consortium Chydenius, University of Jyväskylä, P.O. Box 567, 67701 Kokkola, Finland

*    Correspondence: daranee.lehtonen@tuni.fi

**Abstract:** Despite the increasing use of technologies in the classroom, there are concerns that technology-enhanced learning environments may hinder students' communication and interaction. In this study, we investigated how tangible technologies can enhance students' multimodal communication and interaction during equation-solving pair work compared to working without such technologies. A tangible app for learning equation solving was developed and tested in fourth- and fifth-grade classrooms with two class teachers and 24 students. Video data of the interventions were analysed using deductive and inductive content analysis. Coded data were also quantified for quantitative analysis. Additionally, teacher interview data were used to compare and contrast the findings. The findings showed that the tangible app better promoted students' multimodal communication and peer interaction than working only with paper and pencil. When working in pairs, tangible-app students interacted with one another much more often and in more ways than their paper-and-pencil peers. The implications of this study are discussed in terms of its contributions to research on tangible technologies for learning, educational technology development, and the use of tangibles in classrooms to support students' multimodal communication and peer interaction.

**Keywords:** mathematics classroom; tangible user interface; multimodal communication; peer interaction; computer-supported collaborative learning; equation solving; primary school

## 1. Introduction

Different technologies are increasingly being employed in mathematics classrooms for various purposes, such as supporting teaching and learning, improving learning outcomes, and increasing motivation and enjoyment. The potential benefits of technologies for classroom communication and social interaction have also received attention from education and research communities [1,2]. Despite growing interest, there are still concerns that technology-enhanced learning environments may hinder students' communication and interaction [1,3].

Screen-based technological solutions (e.g., computers and tablets) typically used in the classroom have some constraints that obstruct collaborative activities [2]. While working together in front of a computer or tablet, the students' attention is primarily on the screen, which potentially decreases their communication with each other [4] and their awareness of groupmates [5]. Moreover, screen-based technologies usually allow only single-user keyboard-and-mouse or touchscreen control, thus reducing collaboration among students [1]. Tangible technologies are emerging, with technological solutions offering unique interfaces that allow digital information to be intuitively operated through the manipulation of physical objects [6]. A growing body of research has explored the

possible contributions of tangible technologies to educational contexts, including classroom communication and interaction [5,7–9].

A traditional school mathematics lesson typically consists of teacher-led instruction and students' independent practice, in which students individually and silently complete exercises in their textbooks [10]. Nevertheless, it has been acknowledged that students are likely to learn mathematics better through multimodal (e.g., verbal, visual, mathematical, and gestural) communication [10] and peer interaction [11,12]. When used appropriately, technologies can benefit mathematics education [13,14], but their use in the classroom is still relatively limited [15], particularly for supporting discussion and collaboration [13]. Previous studies (e.g., [14,16]) have demonstrated that technology use within a social context can promote student achievement in mathematics. Therefore, it is important to explore the potentials of technologies for supporting multimodal communication and peer interaction in mathematics classrooms.

To investigate how tangible technologies can enhance students' multimodal communication and interaction, the current study utilised research data collected during the first author's doctoral dissertation [3], in which a tangible app for equation solving was developed and tested in schools. In our previous work [16], we analysed parts of the dissertation data to examine the benefits of tangible technologies in mathematics classrooms with regard to impacts on learning, learning support, and usability. In this study, we largely relied on the dissertation class intervention data to explore the potential role of tangible technologies in students' multimodal communication and interaction. We compared how primary students collaboratively learned to solve linear equations using either the developed tangible app or paper and pencil, a dominant way of working in typical mathematics classrooms [17,18]. In doing so, we attempted to answer the following questions:

1. What are the characteristics of students' multimodal communication and peer interaction during mathematics pair work sessions with and without tangible technologies?
2. How do tangible technologies support students' multimodal communication and peer interaction in learning mathematics?

Our findings show that the tangible app facilitated students' multimodal communication and interaction compared to working only with paper and pencil. This study can benefit researchers interested in tangible technologies in educational contexts, particularly for student communication and interaction. It will assist educational technologists in developing technological solutions for classroom communication and social interaction. It will also encourage practitioners to use emerging technologies in their classrooms to support students' multimodal communication and collaborative interactions.

## 2. Theoretical Background

### 2.1. Learning through Social Interaction

Vygotsky [19] saw learning from the socio-constructivist perspective as a collaborative knowledge-building process in which learning takes place through social interaction. According to his work, each learner functions at a particular level independently and has the potential to attain an upper level of learning capacity under teacher guidance, in collaboration with more advanced peers, or with the support of concrete tools or technology-enhanced learning environments. The difference between these two levels is known as the zone of proximal development (ZPD) [19]. According to Piaget's [20] perspective, an individual learner is more likely to learn by interacting with a peer who is viewed as a reciprocal partner and can provide a conflicting perspective on how to solve a problem. When the conflicting perspective creates an optimal mismatch with the learner's current level of understanding, growth to the upper level of learning capacity is likely to occur [21,22] (cf., [12]).

A collaborative knowledge-building process enables support in a timely manner within a learner's ZPD [19]. In mathematics classrooms, this process can take place through discussions in pairs, in small groups, or with the whole class. Classroom discussions enable students to explain, argue, and justify, for example, a mathematical concept, which

can help students develop their understanding of that concept (e.g., [12,23,24]). From a cognitive perspective, explaining their own mathematical thinking to peers verbally or through pictures, mathematical symbols, or concrete tools [25] helps students remember that information and relate it to prior information in their memory [26].

We can see classroom discussions as a scaffolding process (see [27]) in which a more capable peer or teacher helps a student solve a problem, carry out a task, or achieve a goal that would otherwise be beyond their capacity (cf., learning within the ZPD). When learning mathematics through social interaction, students can help one another develop their procedural skills, conceptual understanding, metacognitive strategies, and mathematical practices [12]. To achieve collaborative knowledge building, students need to communicate well [28]. Different methods of communication can facilitate classroom discussions.

### 2.2. Multimodal Communication in Mathematics Classrooms

Mathematics education has traditionally focused on the use of mathematical symbols to show procedures for arriving at a solution without any accompanying text or drawings. During mathematics lessons, teachers typically explain how to solve difficult math problems and teach new content through speech, mathematical symbols, and/or drawings. In turn, students are mostly passive listeners of the teacher's instruction and silent mathematics exercise workers.

The role of multimodal (i.e., different semiotic systems) communication in mathematics classrooms has recently gained more attention (e.g., [10,29–31]). For example, the Finnish National Core Curriculum for Basic Education [32] has encouraged primary students to communicate their conclusions and solutions to teachers and peers using spoken and written language, drawings, concrete tools, and information and communication technology. Previous studies (e.g., [10,33]) have indicated that multimodal communication in the classroom can help students learn mathematics.

Four semiotic systems or *languages* used for communication in mathematics classrooms can be distinguished: natural (spoken and written), mathematical symbolic (numbers and symbols), pictorial (e.g., pictures and graphs), and tactile functional (e.g., concrete tool manipulation) [25]. Tactile functional language can be expanded to body language, which includes any communication through physical behaviours, such as facial expressions, gestures, touch, motion, and full-body interaction. The process of using these languages to express mathematical thinking is called *languaging* [34], which can be seen as multimodal communication and contributes to mathematics classrooms in three ways [10]. First, languaging helps students organise their own mathematical thinking and develop better understanding. Second, it provides other students with different views on the subject being discussed, which helps them further develop their own thinking. Third, it assists teachers in evaluating students' understanding, which can be used for further support and lesson planning.

### 2.3. Tangible Technologies for Communication and Interaction in the Classroom

Tangible technologies have increasingly been piloted in different educational domains and at different levels. Studies to date have shown that these emerging technologies can benefit educational contexts by adding physical actions to computer-based learning activities [9], concretising abstract to-be-learned content through multimodal mappings [9,35], scaffolding learning and therefore encouraging independent exploration [5,7], increasing engagement and enjoyment [5,7], facilitating multimodal communication [4,5], and promoting collaborative interaction [5,7,9,35].

Unlike screen-based technologies, tangible technologies provide various affordances for face-to-face communication and collaborative interaction [2,5]. First, when gathering around a co-located tangible, students can see one another and each other's actions [9,36], which encourages verbal and non-verbal (e.g., facial expressions, gestures, and full-body interactions) communication [4,5]. Co-located tangibles also promote students' collective exploration and collaborative knowledge building by making each student's input and

the consequent digital output visible to everyone [5]. Second, tangibles facilitate students' collaborative knowledge construction through shared representations of the task [5,36]. Third, multiple students can simultaneously complete a task by manipulating shared resources, such as multiple physical objects [6,35], which encourages everyone to engage in the group activity [5,7]. Finally, working with tangibles leads to parallel actions in which one student's interface operation intervenes in others' current or planned interactions with the interface [5]. Consequently, students need to pay attention to others' actions, negotiate with their groupmates (e.g., by discussing or exchanging objects), and synchronise their own actions, all of which promote their collaborative interaction [5]. To date, tangibles have been used to facilitate collaborative interaction in four different contexts: exploration, problem solving, skill and knowledge development, and communication [2,7].

## 3. Materials and Methods

### 3.1. The Tangible App for Equation Solving

The first author, together with a team of computer science students and their supervisor, developed a prototype of the tangible app for primary students to learn linear equation solving (see [16] for the detailed design, development, and implementation of the app). The design was informed by the literature on to-be-learned content, pedagogy (including learning through peer interaction), and educational technologies as well as our investigation of existing educational solutions and empirical studies [3]. One design consideration of the app was based on our findings that physical learning tools better promote peer interaction than their digital counterparts.

The tangible app is composed of a tablet app and two types of physical objects: *X*-boxes designed to represent unknowns and widely used base-10 blocks representing constants. The placement and removal of physical objects on a digital scale on a tablet screen is detected using image recognition via an external web camera connected to the tablet. The app is divided into two levels: level 1 for equation solving by substituting values for the unknown (Figure 1) and level 2 for equation solving by performing the same operation on both sides of the equation (Figure 2). Each level has eight equations to be modelled and solved.



**Figure 1.** How to complete the level 1 exercise $x + 1 = 4$. (**a**) Equation modelling. (**b**) Equation solving by substituting values for the unknown. (**c**) The digital scale is balanced when the equation is solved (from [16] (p. 10) CC BY-NC).

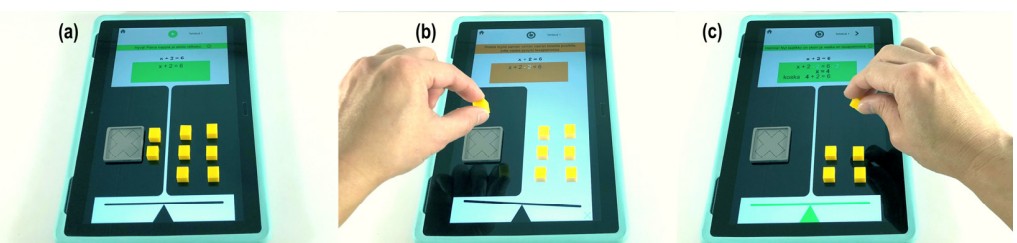

**Figure 2.** How to complete the level 2 exercise $x + 2 = 6$. (**a**) Equation modelling. (**b**) Equation solving by performing the same operation on both sides of the equation. (**c**) The digital scale is balanced when the equation is solved (from [16] (p. 10) CC BY-NC).

Two sets of instructional materials (teacher guides and student worksheets) were also designed to accompany both levels of the app. The teacher guides provide key mathematical concepts that students need to understand to master equation solving, whereas the student worksheets contain the same eight equations as in the app. To complete the worksheets, students are asked to first translate equations, which are presented as a picture, mathematical sentence, or word problem, into two other representations and then solve the equations.

### 3.2. Participants

The equitable school system in Finland [37] made it possible to use convenience sampling in this study, despite a small sample size. Altogether, 12 fourth graders (six girls and six boys, ages 10–11) and 12 fifth graders (four girls and eight boys, ages 11–12) from primary schools in Southern Finland and their class teachers (one female and one male with 10 and 11 years of teaching experience, respectively) willingly participated in class interventions. Students were recruited based on their legal guardians' informed consent and their mathematics performance throughout their school years. The students in both grades had mixed attainment levels (high, medium, and low) in mathematics and received no formal instruction in equation solving prior to the study. Both teachers sometimes used physical and digital teaching materials in their mathematics classrooms. They had some experience in teaching linear equations but not with teaching materials resembling our tangible app.

### 3.3. Research Design and Procedures

The 45 min class interventions took place during school hours in real classroom settings: fourth-grade interventions took place in their own classroom and fifth-grade interventions took place in a small school classroom. Each intervention was divided into two parts: whole-class instruction and a pair work session. During the first 10–15 min, the fourth-grade teacher ($T_1$) taught equation solving to own students ($St_{1-12}$) by substituting values of the unknown and the fifth-grade teacher ($T_2$) taught equation solving to own students ($St_{13-24}$) by performing the same operation on both sides of an equation using the provided teacher guide. Then, as shown in Table 1, each teacher assigned their own students equally into two groups (learning with paper and pencil or the tangible app) and each group into three pairs: high- and medium-attaining (A and B), medium- and medium-attaining (B and B), and medium- and low-attaining (B and C). Students with heterogeneous attainments were paired together to create opportunities for them to collaboratively construct knowledge within their ZPD [38]. Additionally, the teachers took the students' ability to work well together into account when pairing them to ensure peer interaction [38,39].

**Table 1.** Participating student ($N = 24$) pairs by grade, mathematics attainment level, and pair work condition.

| Grade | Pair Work Condition | |
| :---: | :---: | :---: |
| | **Paper-and-Pencil Pair** | **Tangible-App Pair** |
| 4 | $St_1$ (A) and $St_2$ (B)<br>$St_5$ (B) and $St_6$ (B)<br>$St_9$ (B) and $St_{10}$ (C) | $St_3$ (A) and $St_4$ (B)<br>$St_7$ (B) and $St_8$ (B)<br>$St_{11}$ (B) and $St_{12}$ (C) |
| 5 | $St_{13}$ (A) and $St_{14}$ (B)<br>$St_{17}$ (B) and $St_{18}$ (B)<br>$St_{21}$ (B) and $St_{22}$ (C) | $St_{15}$ (A) and $St_{16}$ (B)<br>$St_{19}$ (B) and $St_{20}$ (B)<br>$St_{23}$ (B) and $St_{24}$ (C) |

Note. Each student is denoted with a student number (attainment level). $St_n$, n = 1–24. (A), (B), or (C) = high-, medium-, or low-attaining, respectively.

After the teacher-led instruction, the students worked in pairs under their teachers' supervision for 30–35 min. They had to complete eight exercises on the worksheet by

modelling equations represented through one of three representations into two others and then solving each equation. The paper-and-pencil students were only provided with an individual worksheet, while the tangible-app students received an individual worksheet and a shared tangible app. After the interventions, each teacher participated in a face-to-face semi-structured interview regarding their experiences and opinions about the interventions.

### 3.4. Data Collection and Analysis

Each pair was video-recorded with two cameras (one from the side and one from the back) to ensure well-captured data. In total, 530 min of video data from 12 pair work sessions were recorded. All videos were watched and transcribed by the first author. Due to the background noise in the classrooms and no use of additional microphones, the audio quality of most videos was too poor for transcribing students' dialogues. Therefore, we decided to transcribe only their on-task peer communication topics (modelling and solving equations) and related non-verbal actions. The teacher interviews were audio-recorded and transcribed.

Our analysis focused on how each pair work condition promoted students' multimodal communication and interaction with peers to learn linear equations. We relied primarily on video data and accompanying transcriptions. We also used teacher interview data to compare and contrast our findings.

The video analysis consisted of two parts: (1) the frequencies of students' on-task peer communication (see [16] for more details) and (2) their peer interaction characteristics. For the first part, the video transcription was first analysed using a qualitative deductive content analysis [40] to identify students' on-task peer communication directions (one-way or two-way) and modalities (spoken natural language, body language, or natural language and body language) according to a categorisation matrix [16], which was built based on a theoretical framework. After that, individual actions were combined into a communication episode, which is a unit of complete actions for one specific objective. For example, one communication episode of modelling an equation might consist of a pair's discussion about how to model an equation and their whole process of modelling the equation together. Then, all discovered communication episodes were quantified (i.e., counted) and categorised into specific communication directions and modalities for descriptive statistical analysis. Students' off-task peer communication (e.g., small talk or how to operate the app) and any communication with the teacher were not part of this analysis. A Pearson's chi-squared test was used to statistically investigate the relationships between pair work conditions and peer communication directions and modalities. For the second part, the video transcription was analysed using a qualitative inductive content analysis (through open coding, creating categories, and abstraction [40]) to identify certain patterns of how the students in each condition interacted with their groupmates.

## 4. Results

To investigate how tangible technologies support students' peer interaction when learning linear equations in pairs, we compared the students' pair work sessions using our tangible app to the sessions that did not. Next, we provide descriptive statistics of students' peer communication, their peer interaction analysis, and teachers' observations of their students' pair work.

### 4.1. Descriptive Statistics

Altogether, 287 episodes of on-task peer communication were observed during 12 pair work sessions (6 sessions for each condition). Most of the communication occurred during the tangible-app sessions (70.4%). In completing the eight exercises, the average peer communication of the tangible-app students (34 episodes/pair, SD = 13.9) was more than twice that of the paper-and-pencil students (14 episodes/pair, SD = 7.7). Table 2 shows the frequencies and percentages of students' on-task communication episodes that occurred during each pair work condition regarding communication directions and modalities.

A chi-squared test showed that there were associations between pair work conditions and peer communication directions ($X2$ (1, N = 287) = 23.15, $p < 0.001$) and modalities ($X2$ (2, N = 287) = 17.13, $p < 0.001$).

**Table 2.** Frequencies and percentages of on-task peer communication episodes (*N* = 287) regarding directions and modalities by pair work condition.

| | Pair Work Condition | | | |
| | Paper-and-Pencil | | Tangible App | |
| **Communication** | *n* | % | *n* | % |
|---|---|---|---|---|
| One-Way Communication | | | | |
| Spoken natural language | 40 | 13.9 | 55 | 19.2 |
| Body language | 7 | 2.4 | 5 | 1.7 |
| Spoken natural language and body language | 4 | 1.4 | 0 | 0 |
| Two-Way Communication | | | | |
| Spoken natural language | 22 | 7.7 | 42 | 14.6 |
| Body language | 0 | 0 | 11 | 3.8 |
| Spoken natural language and body language | 12 | 4.2 | 89 | 31.0 |
| Total | 85 | 29.6 | 202 | 70.4 |

We further examined two-way communication (sending and receiving information) episodes occurring during each pair work condition to indicate students' peer interactions. Tangible-app students (70.1% of their communication) interacted with their groupmate much more than paper-and-pencil students (40.0% of their communication). Paper-and-pencil students mostly interacted with each other through spoken natural language (64.7% of their two-way communication), while tangible-app students did so through spoken natural language and body language (62.7% of their two-way communication), for example, manipulating the app and simultaneously talking about their own actions. Sometimes, students in the tangible-app condition also interacted with each other only by manipulating the app to complete an exercise without talking to their groupmates.

### 4.2. Analysis of Peer Interaction

The findings from the qualitative analysis of the video data were consistent with the quantitative analysis. Peer interaction was typically observed during tangible-app sessions, while silent and individual work was usually observed during paper-and-pencil sessions. In the following subsections, we describe how the students in each condition worked in pairs, with accompanying examples of video transcription, which were translated into English.

#### 4.2.1. Paper-and-Pencil Condition

Without the teacher's presence, paper-and-pencil students rarely interacted with their groupmates. They usually worked quietly on their individual worksheets at different paces. The following example, where fifth-grade medium- and low-attaining students modelled and solved an equation represented as a picture, illustrates typical pair work of students in this condition (Figure 3).

Students $St_{21}$ (B) and $St_{22}$ (C) completing exercise 5 (1:00 min):

| | |
|---|---|
| $T_2$: | leaves after Exercise 4. |
| $St_{21}$ (B): | silently models the given picture by writing an equation (a mathematical sentence representing the picture) on her own worksheet. |
| $St_{22}$ (C): | silently models the given picture by writing an equation (a mathematical sentence representing the picture) on her own worksheet. |
| $St_{21}$ (B): | silently solves the equation by writing mathematical symbols on her own worksheet. |
| $St_{22}$ (C): | silently solves the equation by writing mathematical symbols on her own worksheet. |

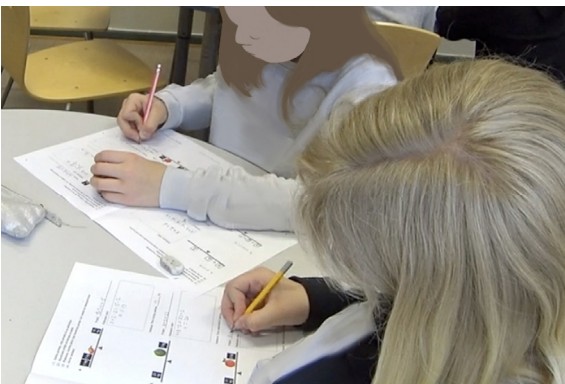

**Figure 3.** No peer interaction. students $St_{21}$ (B) and $St_{22}$ (C), in the paper-and-pencil group, working on their own worksheets quietly and separately.

Students in this condition occasionally sought help from their groupmates by asking or looking at what they had written on their worksheets. However, they seldom had their groupmate's attention. The teacher's encouragement to work together appeared to promote peer scaffolding among paper-and-pencil students. With the encouragement of teachers, higher-attaining students verbally advised their groupmates, who then followed the given advice. In the following example, fourth-grade medium- and low-attaining students modelled and solved an equation represented as mathematical symbols, first independently and then with the teacher's guidance.

Students $St_9$ (B) and $St_{10}$ (C) completing exercise 1 (5:15 min):

| | |
|---|---|
| $St_9$ (B): | silently models the given equation by drawing pictures on his own worksheet. |
| $St_{10}$ (C): | silently reads his own worksheet, then says something to $St_9$ (B) while pointing at his groupmate's worksheet (Figure 4a). |
| $St_9$ (B): | does not react to what $St_{10}$ (C) did and continues solving the equation by writing mathematical symbols on his own worksheet. |
| $T_1$: | comes to explain to the students how to model the given equation by drawing. |
| $St_{10}$ (C): | talks to $T_1$ about drawing the equation. |
| $St_9$ (B): | finishes exercise 1 on his own worksheet and starts completing exercise 2. |
| $T_1$: | asks $St_9$ (B) to explain to $St_{10}$ (C) how he drew exercise 1. |
| $St_9$ (B): | replies to $T_1$, then explains to $St_{10}$ (C) how to draw the equation while pointing at $St_{10}$ (C)'s worksheet (Figure 4b). |
| $St_{10}$ (C): | listens to $St_9$ (B), then draws the equation on his own worksheet. |
| $St_9$ (B): | continues working on exercise 2. |
| $St_{10}$ (C): | silently solves exercise 1 by writing mathematical symbols on his own worksheet. |
| $St_9$ (B): | finishes exercise 2 on his own worksheet and starts completing exercise 3 without waiting for $St_{10}$ (C). |

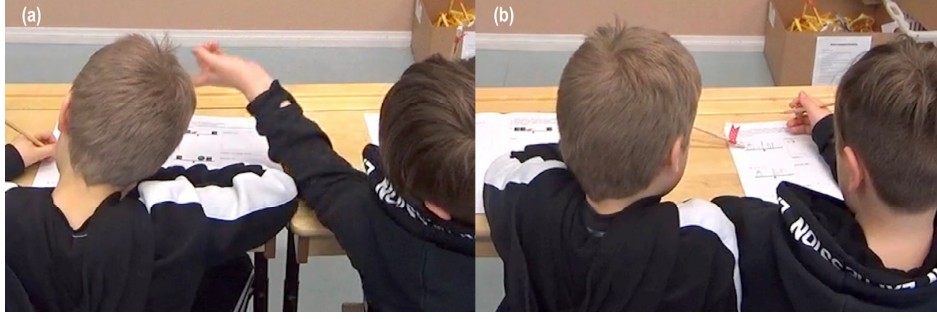

**Figure 4.** Peer communication and scaffolding. (**a**) $St_{10}$ (C) (on the right) asking for help from $St_9$ (B) (on the left) and pointing at his groupmate's worksheet but receiving no reaction from $St_9$ (B). (**b**) Later, $St_9$ (B) explaining to $St_{10}$ (C) by speaking and pointing at his groupmate's worksheet.

Sometimes, even the teacher's encouragement to work together could not make paper-and-pencil students interact with each other. Instead of responding to the teacher's encouragement, students sometimes continued working separately. A temporary lack of concentration, such as looking at what other pairs were doing, was occasionally observed. In the following example, fourth-grade high- and medium-attaining students separately and silently worked on an equation represented as mathematical symbols, even though their teacher had encouraged them to talk to each other. At some point, the high-attaining student lost his concentration and turned to look at what another pair was doing.

Students $St_1$ (A) and $St_2$ (B) completing exercise 2 (1:10 min):

| | |
|---|---|
| $St_1$ (A): | silently models the given equation by drawing on his own worksheet. |
| $St_2$ (B): | silently models the given equation by drawing on her own worksheet. |
| $T_1$: | comes and asks $St_1$ (A) and $St_2$ (B) to talk to each other about how to solve the equation. |
| $St_1$ (A): | does not respond to the teacher's request and starts to solve the equation by writing mathematical symbols on his own worksheet. |
| $St_2$ (B): | does not respond to the teacher request and starts to solve the equation by writing Finnish texts and mathematical symbols on her own worksheet. |
| $St_1$ (A): | Finishes exercise 2 on his own worksheet and starts completing exercise 3 without waiting for $St_2$ (B). Stops solving his worksheet and turns to look at what another pair is doing (Figure 5), then turns back to his own worksheet. |

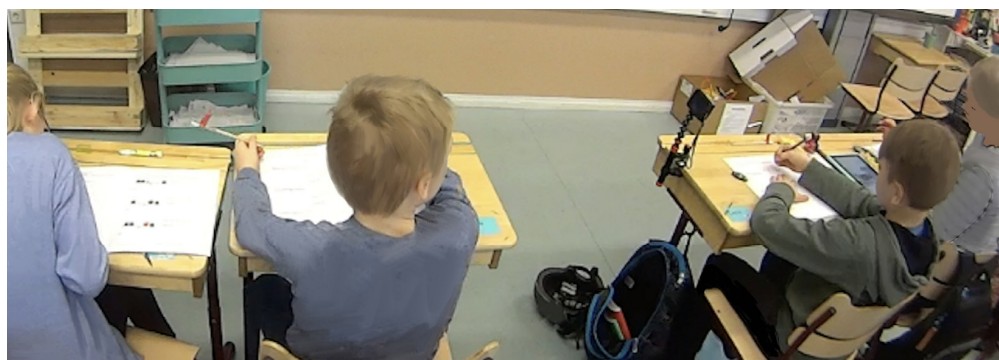

**Figure 5.** Lack of concentration. $St_1$ (A) (second from the left) turning from his worksheet to look at what the tangible-app pair is doing (on the right).

### 4.2.2. Tangible-App Condition

Students in the tangible-app condition typically interacted with their groupmates, even without the teacher's encouragement. Before using the app to model and solve equations, students usually discussed what to do. Then, they completed the exercises together by manipulating physical objects on the tablet while describing what they were doing. Occasionally, they manipulated objects without saying anything. Students' peer interaction during their equation modelling and solving was either scaffolding or collaboration. After the equation had been modelled and solved, the students individually recorded their work on their own worksheets.

Two types of peer scaffolding were often observed. The first was advice giving and taking. In this type of scaffolding, higher-attaining students told their groupmates how to model or solve the given equation using the app. Then, when their groupmates used the app according to the given advice, the advice givers usually monitored or whispered to ensure that everything went well. In the following example, a fifth-grade medium-attaining student assisted his low-attaining groupmate in using the app to model and solve an equation represented as a picture.

Students $St_{23}$ (B) and $St_{24}$ (C) completing Exercise 5 (2:10 min):

| | |
|---|---|
| Both students: | read their own worksheets then look at the tablet. |
| St$_{24}$ (C): | asks St$_{23}$ (B) how to model the given picture in the worksheet using the app. |
| St$_{23}$ (B): | explains to St$_{24}$ (C) what to do. |
| St$_{24}$ (C): | places physical objects on both sides of the digital scale on the tablet screen to model the equation according to St$_{23}$ (B)'s advice. |
| St$_{23}$ (B): | observes St$_{24}$ (C) modelling the equation using the app. |
| Both students: | write a mathematical sentence of the equation, which St$_{24}$ (C) has just modelled on their own worksheet, then look at the tablet. |
| St$_{24}$ (C): | asks St$_{23}$ (B) how to solve the equation using the app. |
| St$_{23}$ (B): | tells St$_{24}$ (C) what to do while pointing at the app (Figure 6a). |
| St$_{24}$ (C): | removes base-10 blocks from both sides of the digital scale to solve the equation according to St$_{23}$ (B)'s advice. |
| St$_{23}$ (B): | observes St$_{24}$ (C) solving the equation using the app until the end (Figure 6b). |
| Both students: | look at the mathematical sentences of the equation-solving process on the tablet screen and write the equation-solving process as Finnish texts and mathematical symbols on their own worksheets. |

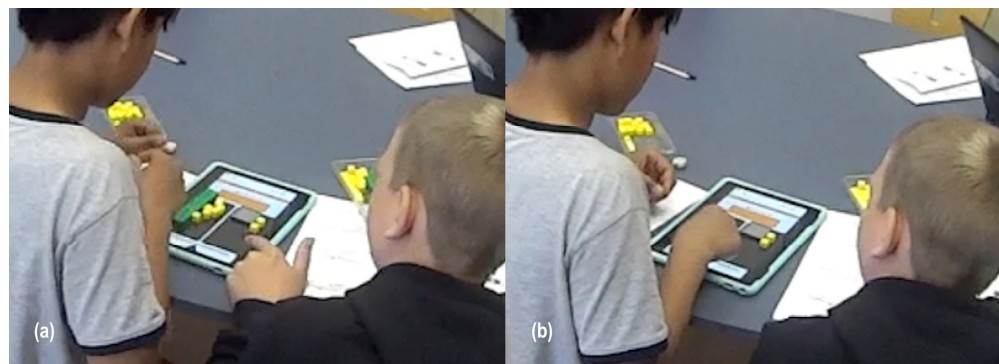

**Figure 6.** Peer scaffolding. (**a**) St$_{23}$ (B) (on the right) explaining to St$_{24}$ (C) (on the left) how to solve the equation. (**b**) St$_{24}$ (C) removing base-10 blocks from the digital scale to solve the equation, while St$_{23}$ (B) observes what St$_{24}$ (C) is doing.

The second type of scaffolding was demonstrating and watching. In this type of scaffolding, the higher-attaining students showed their groupmates how to model or solve the given equation by manipulating objects on the tablet and usually talking at the same time. Lower-attaining students then closely watched what their groupmates were doing. In the following example, a fifth-grade high-attaining student showed her medium-attaining groupmate how to use the app to solve an equation represented as a picture.

Students St$_{15}$ (A) and St$_{16}$ (B) completing exercise 4 (3:10 min):

| | |
|---|---|
| Both students: | read their own worksheets, then look at the tablet and discuss how to model the given picture in the worksheet using the app. |
| St$_{15}$ (A): | places physical objects on both sides of the digital scale on the tablet screen to model the equation while describing what she is doing. |
| St$_{16}$ (B): | simultaneously places physical objects on both side of the digital scale on the tablet screen to model the equation while describing what she is doing. |
| Both students: | discuss how to solve the equation using the app. |
| St$_{15}$ (A): | removes base-10 blocks from both sides of the digital scale to solve the equation while describing what she is doing. |
| St$_{16}$ (B): | watches St$_{15}$ (A) solving the equation using the app until the end (Figure 7). |
| Both students: | look at the mathematical sentences of the equation-solving process on the tablet screen and write the equation-solving process as Finnish texts and mathematical symbols on their own worksheets. |

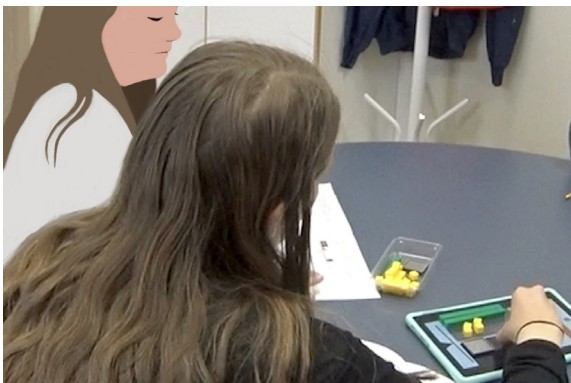

**Figure 7.** Peer scaffolding. St$_{15}$ (A) (in front) showing and telling St$_{16}$ (B) (in back) how to solve the equation with the app while St$_{16}$ (B) is watching what St$_{15}$ (A) is doing.

Two types of collaboration, doing together and turn taking, were typically observed in the tangible-app condition. In the first type of collaboration, students used the app to model or solve the given equation together. Both students simultaneously placed physical objects on or removed them from the tablet screen. In the second type of collaboration, students took turns using the app to model or solve the given equation. They manipulated physical objects on the tablet one at a time and watched while their groupmates manipulated objects. All students in this condition were engaged in using the app to complete all the exercises until the end. No off-task activities were observed during their pair work sessions. Some students also expressed their enjoyment in their pairs' success in using the app to model or solve the given equation. They smiled, laughed, raised hands, or said, "Yeah!". In the following example, fourth-grade high- and medium-attaining students first simultaneously used the app to model an equation represented as a picture and then took turns solving the equation. After the completion of the exercise, the medium-attaining student showed a sign of enjoyment.

Students St$_3$ (A) and St$_4$ (B) completing exercise 3 (1:30 min):

| | |
|---|---|
| Both students: | read their own worksheets, then look at the tablet and discuss how to model the given picture in the worksheet using the app. |
| St$_3$ (A): | places physical objects on the left side of the digital scale, which is close to him, on the tablet screen to model the left side of the equation while talking about what he is doing. |
| St$_4$ (B): | simultaneously places physical objects on the right side of the digital scale, which is close to her, on the tablet screen to model the right side of the equation while describing what she is doing (Figure 8a). |
| Both students: | discuss how to solve the equation using the app. |
| St$_3$ (A): | adds some base-10 blocks to the *X*-box on the left side of the digital scale to solve the equation. |
| St$_4$ (B): | waits until St$_3$ (A) stops, then adds more base-10 blocks to the *X*-box to solve the equation (Figure 8b). |
| St$_3$ (A): | Waits until St$_4$ (B) stops, then adds more base-10 blocks to the *X*-box to finish solving the equation (Figure 8c). The equation is solved. The digital scale is balanced and turns green with a ding sound. |
| St$_4$ (B): | says "Yeah!" and raises her hands into the air. |
| Both students: | draw pictures representing the given equation and write the solution on their own worksheets. |

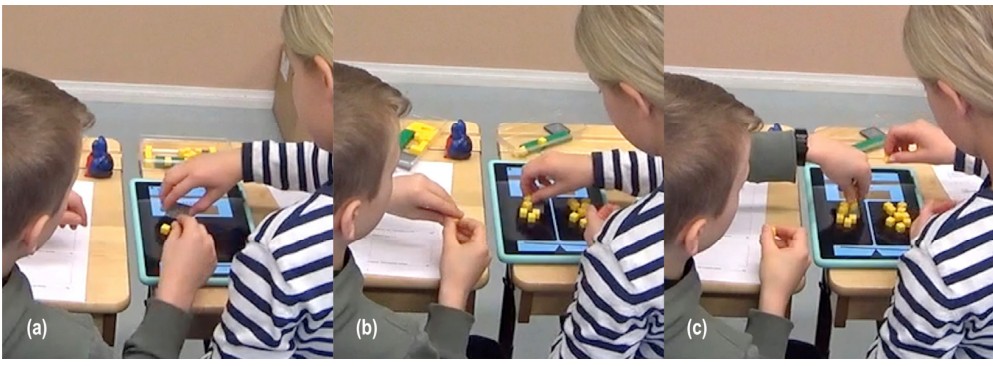

**Figure 8.** Peer collaboration. (**a**) St$_3$ (A) (on the left) and St$_4$ (B) (on the right) simultaneously adding physical objects to the digital scale to model the equation. (**b**) Later, St$_4$ (B) adding another base-10 block to the *X*-box to solve the equation, while St$_3$(A) watches. (**c**) St$_3$ (A) adding the last base-10 block to the *X*-box to solve the equation, while St$_4$ (B) watches.

Occasionally, pair work was dominated by one student. In this case, a dominant student manipulated the app too quickly for their groupmate to keep pace. Sometimes, it was possible for the groupmate to take part in the app manipulation using available physical objects. When it was impossible, the groupmate could only watch what the other was doing or count objects that were placed on or removed from the tablet screen. In the following example, two fourth-grade medium-attaining students used the app to model an equation given as a word problem. One student began to solve the equation without letting his groupmate participate in the app manipulation.

Students St$_7$ (B) and St$_8$ (B) completing exercise 7 (4:30 min):

| | |
|---|---|
| T$_1$: | asks both students to read the word problem in the worksheet, then guides them on how to translate the word problem into a mathematical sentence representing the equation. |
| St$_7$ (B): | nods. |
| T$_1$: | asks St$_7$ (B) to explain what he understands to St$_8$ (B). |
| St$_7$ (B): | explains it to St$_8$ (B). |
| Both students: | write the mathematical sentence and draw pictures representing the word problem on their own worksheets, then look at the tablet. |
| T$_1$: | leaves. |
| St$_8$ (B): | places physical objects on the right side of the digital scale, which is close to him, on the tablet screen to model the right side of the equation while describing what he is doing. |
| St$_7$ (B): | after St$_8$ (B) has finished placing physical objects on the right side of the scale, places physical objects on the left side of the digital scale, which is close to him, on the tablet screen to model the left side of the equation while talking about what he is doing. |
| St$_8$ (B): | tries to add base-10 blocks to the *X*-box on the tablet screen to solve the equation. |
| St$_7$ (B): | pushes his teammate's hand away from the tablet (Figure 9a), then quickly adds base-10 blocks to the *X*-box to solve the equation. |
| St$_8$ (B): | tries to add base-10 blocks to the *X*-box but cannot keep pace with St$_7$ (B) (Figure 9b). |
| Both students: | discuss the solution and write it on their own worksheets. |

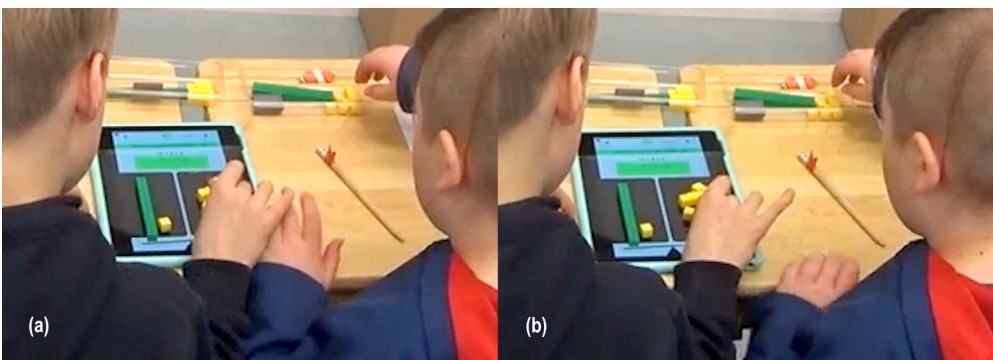

**Figure 9.** Dominance. (**a**) St$_7$ (B) (on the left) pushing St$_8$ (B)'s (on the right) hand away from the tablet. (**b**) St$_7$ (B) quickly adding base-10 blocks to the *X*-box to solve the equation, making it impossible for St$_8$ (B) to join.

*4.3. Teachers' Observations of the Interventions*

Both teachers made observations that reflected the findings from the video transcription. They noticed that during the pair work sessions, paper-and-pencil students tended to work individually on their own worksheets without interacting with their groupmates. The teachers had to frequently remind paper-and-pencil students to complete the exercises together and talk to each other. The fifth-grade teacher thought that the worksheet encouraged students to start filling it out. The fourth-grade teacher argued that because her students were unfamiliar with verbalising their mathematical thinking, it was easier for paper-and-pencil students to write down their solutions on their own worksheets.

Both teachers noticed that tangible-app students actively worked in pairs without the teacher's encouragement and concentrated on solving equations until the end. They thought that the app better promoted students' peer interaction than working with paper and pencil. The fourth-grade teacher felt that this could be because students had to manipulate objects on the shared tablet and were therefore less tempted to write their solution on their own worksheet straight away. In the fifth-grade teacher's opinion, the app manipulation encouraged students to think aloud and discuss what objects to place on or remove from each side of the digital scale. However, the fourth-grade teacher noted that while the app encouraged students to express their mathematical thinking through physical actions, it could discourage their verbalisation. Thus, students might only manipulate objects without talking to each other.

## 5. Discussion

The purpose of this study was to explore the potential of tangible technologies for students' multimodal communication and interaction during equation-solving pair work sessions. In Sections 5.1 and 5.2, we discuss the findings gained from video data of the interventions with and without tangible technologies as well as teacher interviews to address the first and second research questions, respectively. We also explore the implications of the findings. In Section 5.3, we reflect on the limitations of our study and suggest directions for future research.

*5.1. Multimodal Communication and Peer Interaction with and without Tangible Technologies*

In contrast to concerns that technologies may hinder students' communication and interaction, we found that when working in pairs, tangible-app students interacted with one another (two-way communication) much more often and in more ways than their paper-and-pencil peers. The associations between pair work conditions and students' communication frequencies and modalities were statistically significant. Our findings support the results of a meta-analysis of studies between 2007 and 2021 on tangible technologies for learning [7], which showed that tangibles can encourage students to work together instead of working alone.

Despite teachers' encouragement to complete the exercises together, paper-and-pencil students mainly worked on their own worksheets with minimal interaction between groupmates, as in traditional mathematics classrooms. When communicating with one another, they mainly used spoken natural language. Similar to the findings of [41,42], we found that tangibles contribute to students' peer interactions in mathematics classrooms. Tangible-app students collaboratively learned to solve equations without the teacher's encouragement.

Various characteristics of successful collaborative interaction proposed by [43] were observed during the tangible-app pair work sessions. There was evidence of students' mutual understanding: students used different means (spoken natural language and body language) to send information to their groupmates, and groupmates usually provided verbal or non-verbal feedback of their understanding. Students usually managed their smooth interactions by taking turns or collaboratively manipulating the app. They also discussed how to model or solve the equation and then either made a joint decision or more capable peers gave advice. Students demonstrated good interpersonal relationships: they typically completed the exercises either as partners or as tutor and tutee, and solo behaviours were occasionally observed. In contrast to their paper-and-pencil peers, tangible-app students focused on completing the exercises in pairs until the end and often showed their enjoyment in accomplishing the task, which indicated their motivation and commitment to it.

### 5.2. Tangible Technologies' Support for Multimodal Communication and Peer Interaction

Our findings suggest that tangibles better promote students' multimodal communication and peer interaction than working only with paper and pencil. We found five possible reasons that may have contributed to this (cf., [7]). First, the app was a co-located tangible, which positioned students to face one another and made the app input and output visible to all students. When manipulating physical objects on the tablet between them, the students had to look at one another and see what the other was doing [9,36]. Awareness of the current state of peers is important for successful collaborative activities [28]. Students could easily observe how the other student manipulated physical objects and the consequent digital output [5], which reinforced the role of each student to ensure a proper contribution to the pair work [28]. In contrast, paper-and-pencil students focused more on completing their individual worksheets and hardly noticed what their groupmates were doing. Moreover, it was difficult to actually see what the other was doing because their action (writing on their own worksheet) was less observable than physical object manipulation.

Second, the tangible-app students shared a tablet, which enabled them to have a shared representation of the task, a common goal, and a joint reward. Students could see equations presented physically and digitally on the shared tablet, which helped them construct their knowledge together (e.g., to-be-solved equations and equation-solving processes) [5,36]. Equations presented on the shared tablet acted as the students' common goal to collaboratively model and solve the equations [28]. Moreover, after an equation was solved, all students received joint rewards (the digital scale was balanced and turned green with a ding sound), which encouraged their engagement and collaboration [28].

Third, the physical and digital affordances of the app facilitated students' multimodal communication by enabling them to communicate with one another through different semiotic systems. We found similar findings as in our previous research [44]: when manipulating physical objects, students were likely to think aloud. When discussing or explaining to their groupmates how to model or solve an equation, students were able to talk while simultaneously pointing at physical objects on the digital balance or placing/removing them. Similar to findings from [45], students who had difficulty expressing their thinking in words [46] or writing [47] could communicate with their groupmates by manipulating physical objects without saying anything. Moreover, it was easier, particularly for students at this age, to observe while peers were manipulating physical objects than when they were writing solutions on their own worksheets.

Fourth, the tangible app provided physical objects, which allowed multiple students to use them simultaneously to model and solve equations [6,35]. Our findings support a study [41] that found that shared input resources balanced students' participation in collaborative activities by encouraging every student to take part [5,7]. When one student tried to take over the entire operation of the app, the other student was still usually able to manipulate objects close to them.

Finally, corresponding to the explanation in [5] about working with tangibles and parallel actions, students needed to monitor how their groupmates manipulated physical objects so they could adjust their own actions accordingly. For example, both students grasped three base-10 blocks from the desk and intended to add them to the *X*-box to balance the digital scale. However, one student was faster and added two blocks to the *X*-box. The other student noticed that, so they only added one block.

To conclude, this study not only contributes to research but also to practice. It advances knowledge of how tangible technologies can facilitate students' multimodal communication and peer interaction. It assists educational technologists in developing emerging technological solutions for classrooms. This study also has practical implications. It encourages practitioners to employ tangible technologies in their classrooms to promote students' multimodal communication and collaborative interaction, as encouraged by, for example, the Finnish Core Curriculum for Basic Education [32]. Tangible technologies can also be used to support physical actions in computer-based learning activities, the understanding of abstract contents, scaffolding learning and independent exploration, and learning engagement and enjoyment. The physical and digital attributes of tangible technologies can play an important role in mathematics classrooms, where students, particularly in higher grades, may perceive the use of traditional concrete tools as childish [48]. While the physicality of tangible technologies concretises abstract mathematical concepts, digitality can engage students who are in favour of digital technology in learning mathematics with tangible technologies. In addition, teacher education and professional development should prepare pre- and in-service teachers for the pedagogical incorporation of technologies in their mathematics classrooms.

### 5.3. Limitations and Future Research

The current study has some limitations. First, the small convenience sample size of participants makes it difficult to generalise the research findings. Moreover, the 45 min class intervention was relatively short. Larger sample sizes, possibly randomised, and a longer period of intervention could be helpful in the future to promote research validity. Second, researcher triangulation could not be met since the data were analysed by only one researcher. In future studies, more researchers should be involved in data analysis to increase research reliability. Third, the design of this research poses the question of whether the research findings were partly influenced by the fact that the tangible-app students had something (a tablet) to share, but the paper-and-pencil students did not. Future studies should employ two control groups, one sharing a worksheet and one sharing a worksheet and a non-digital learning tool, such as a physical balance scale. Finally, to further explore the potentials of tangible technologies for mathematics education, future research could investigate their benefits for other mathematics contents and different educational levels.

**Author Contributions:** Conceptualisation, methodology, investigation, formal analysis, data curation, writing—original draft, and visualisation, D.L.; supervision and writing—original draft and review, J.J. and P.P. All authors have read and agreed to the published version of the manuscript.

**Funding:** This research received no external funding.

**Institutional Review Board Statement:** Not applicable.

**Informed Consent Statement:** Prior to the study, informed consent was obtained from the teachers and legal guardians of all students involved.

**Data Availability Statement:** All of the data is contained within the article.

**Acknowledgments:** The authors would like to acknowledge the contributions of Fouzia Khan, Juho Korkala, Roni Perälä, Niko Sainio, and Krishna Bagale for software development under the supervision of Pekka Mäkiaho; Tapio Lehtonen for the UI; and Kari Kouhia for the prototyping. The authors appreciate the teachers and students who participated in the research.

**Conflicts of Interest:** The authors declare no conflict of interest.

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
