# Peer review of "Multimodal Communication and Peer Interaction during Equation-Solving Sessions with and without Tangible Technologies"

_mti, doi:10.3390/mti7010006_

Round 1
Reviewer 1 Report
This is a very important study on a highly significant topic. We have talked about technologies and their applications in teaching but we have not paid enough attention to the issue of pedagogy related to the use of technologies. For this reason, I welcome the study enthusiastically. It is really exciting to see a study on interaction in a math class. Nevertheless, I would like to suggest the following issues for the authors to address through revision:
1. on page 2, the authors refer to 'dissertation data'. What dissertation data? Were the data from this study also from the dissertation?
2. Can the authors briefly mention why it is important to explore social interaction in math class? I think that the argument is there. Traditional math classses did not encourage such interaction.
3. Can the authors briefly mention how the participants were sampled and recruited for participation?
4. When reading the findings, I feel that the authors need to be more explicit about which findings answer which research questions. For this reason, the findings should be presented as answers to the research questions.
5. Can the authors spell out a couple of specific practices that readers can adopt based on the findings?
This is an excellent paper and I do hope that the comments may help the authors further refine their work.
Author Response
We would like to thank the reviewer for kind and constructive comments, which have helped us refine our manuscript. We have incorporated the reviewers’ suggestions, and these changes have been marked up within the manuscript using the ‘Track Changes’ function. Please find below, in red, a point-by-point response to the comments and concerns of the reviewers. All the line numbers refer to the revised manuscript file with tracked changes.
Point 1: On page 2, the authors refer to 'dissertation data'. What dissertation data? Were the data from
this study also from the dissertation?
Response 1: We apologise for not being clear enough regarding this. The data in the current study were collected during the first author’s dissertation. To state this more clearly, we have merged two paragraphs on Page 2 together and added the phrase ‘the current study’ to Line 61 (To investigate how tangible technologies can enhance students’ multimodal communication and interaction, the current study utilised research data collected during the first author’s doctoral dissertation [3],…) and the phrase ‘the dissertation’ to Line 66 (In this study, we largely relied on the dissertation class intervention data to explore the potential role of tangible technologies in students’ multimodal communication and interaction.)
Point 2: Can the authors briefly mention why it is important to explore social interaction in math class? I think that the argument is there. Traditional math classses did not encourage such interaction.
Response 2: We have elaborated this by adding phrases and new sentences to the paragraph on Page 2 (Lines 49–53 and 55–59) as follows:
A traditional school mathematics lesson typically consists of teacher-led instruction and students’ independent practice, in which students individually and silently do exercises in their textbook [10]. Nevertheless, it has been acknowledged that students are likely to learn mathematics better through multimodal (e.g., verbal, visual, mathematical, and gestural) communication [10] and peer interaction [11,12]…Previous studies (e.g., [14,16]) have demonstrated that technology use within the social context can promote student achievement in mathematics. Therefore, it is important to explore the potentials of technologies for supporting multimodal communication and peer interaction in mathematics classrooms.
Point 3: Can the authors briefly mention how the participants were sampled and recruited for
participation?
Response 3: We have elaborated this by adding the sentence ‘Equitable school system in Finland [37] made it possible to use convenience sampling in this study despite a small sample size.’ at the beginning of Section 3.2 Participants (Line 199–200) and the word ‘willingly’ to the second sentence of the paragraph (Line 200–203): ‘Altogether, 12 fourth graders (six girls and six boys, ages 10–11) and 12 fifth graders (four girls and eight boys, ages 11–12) from primary schools in Southern Finland and their class teachers (one female and one male with 10 and 11 years of teaching experience, respectively) willingly participated in class interventions.)’
Point 4: When reading the findings, I feel that the authors need to be more explicit about which findings answer which research questions. For this reason, the findings should be presented as answers to the research questions.
Response 4: We have not made any changes to Section 4 Results, because we believe that it is clearer to present our findings in this way. However, in Section 5 Discussion, we have discussed our findings in Sections 5.1 and 5.2 to address each research question as suggested by the reviewer. We have also rewritten the section introduction to state this more clearly (Lines 544–546): ‘In Sections 5.1 and 5.2, we discuss the findings gained from video data of the interventions with and without tangible technologies as well as teacher interviews to address the first and second research questions, respectively.’
Point 5: Can the authors spell out a couple of specific practices that readers can adopt based on the
findings?
Response 5: In our original manuscript, we have already provided some practical implications in the paragraph before Section 5.3. We have added one more implication to the revised manuscript (Lines 639–641): ‘In addition, teacher education and professional development should prepare pre- and in-service teachers for pedagogical incorporation of technologies in their mathematics classrooms.’
Reviewer 2 Report
Thank you very much for giving me the opportunity to review the paper. The paper "Multimodal Communication and Peer Interaction during Equation-Solving Sessions with and without Tangible Technologies” is interesting.
Describe the methods and materials that need to be referenced in the statistical data processing in greater detail.
Describing experiments, game systems and results is necessary before analyzing them.
Extending the conclusions – what are the additional topics required for review?
Author Response
We would like to thank the reviewer for constructive comments, which have helped us refine our manuscript. We have incorporated the reviewers’ suggestions, and these changes have been marked up within the manuscript using the ‘Track Changes’ function. Please find below, in red, a point-by-point response to the comments and concerns of the reviewers. All the line numbers refer to the revised manuscript file with tracked changes.
Points 1 and 2: Describe the methods and materials that need to be referenced in the statistical data processing in greater detail. Describing experiments, game systems and results is necessary before analyzing them.
Responses 1 and 2: We have elaborated our methods, materials, and experiments as recommended by the reviewer, for example:
- We have added the sentence ‘Equitable school system in Finland [37] made it possible to use convenience sampling in this study despite a small sample size.’ to the beginning of Section 3.2 Participants (Lines 199–200).
- We have added phrases to the first sentence in the last paragraph of Section 3.3 Research Design and Procedures (Lines 230–231): ‘After the teacher-led instruction, the students worked in pairs under their teachers’ supervision for 30–35 minutes. ’
- We have rewritten parts of the last paragraph of Section 3.4 Data Collection and Analysis (Lines 258–266):
…For example, one communication episode of modelling an equation might consist of a pair’s discussion about how to model an equation and their whole process of modelling the equation together. Then, all discovered communication episodes were quantified (i.e., counted) and categorised into specific communication directions and modalities for descriptive statistical analysis. Students’ off-task peer communication (e.g., small talk or how to operate the app) and any communication with the teacher were not part of this analysis. A Pearson’s chi-squared test was used to statistically investigate the relationship between pair work conditions and peer communication directions and modalities…
Point 3: Extending the conclusions – what are the additional topics required for review?
Response 3: We are not sure, whether we have understood the reviewer’s comment correctly. To our understanding, we have added the sentence ‘Finally, to further explore the potentials of tangible technologies for mathematics education, future research could investigate their benefits for other mathematics contents and different educational levels.’ to the end of Section 5.3 Limitations and Future Research.
Reviewer 3 Report
Overall, I thoroughly enjoyed reading this very clearly written paper. I thought this was a very well-crafted study, and adds to what we know about augmenting mathematics instruction with technology. All aspects of the paper were undertaken to a high standard, and I commend the authors for their excellent scholarship.
I have only two minor suggestions for the authors to consider:
1. It may be worth making it more explicit somewhere in your literature review that there is evidence that teachers have tended not to incorporate technology into the mathematics classroom even when it is simply being used to augment traditional instruction, let alone embrace it as a primary mode of learning. For example, Russo et al. (2021) asked 248 Australian primary school teachers to describe their favourite mathematical game for supporting mathematics learning in their classroom. They found that only 4% of teachers described a game involving students or the teacher interacting with digital technology in any capacity (e.g., supportive software, calculator, interactive number chart) and only 1% selected a digital game specifically. Russo, J., Bragg, L. A., & Russo, T. (2021). How Primary Teachers Use Games to Support Their Teaching of Mathematics. International Electronic Journal of Elementary Education, 13(4), 407-419. You may also wish to consult the work of Catherine Attard, for example, Attard, C. (2015). Introducing iPads into primary mathematics classrooms: Teachers’ experiences and pedagogies. In Integrating Touch-Enabled and Mobile Devices into Contemporary Mathematics Education (pp. 193–213). Orlando, J., & Attard, C. (2016). Digital natives come of age: The reality of today’s early career teachers using mobile devices to teach mathematics. Mathematics Education Research Journal, 28(1), 107–121. Of course, the author may have many other such studies they wish to draw upon – the important thing is that the point is made somewhere I think.
2. The other change I think that needs to be made is to the limitations, and possible directions for future research. Rather than conclude: “in the future, paper-and-pencil students should share, for example, a worksheet” I think a stronger comparison would be “a shared worksheet, combined with non-digital physical manipulatives, such as a balance beam, counters and a whiteboard to support collaborative work”. This would be a stronger comparison in my opinion, and gets at what is afforded in particular by the fact that the intervention is digital, whereas most classrooms currently do not attempt to incorporate digital technology (as I noted in my previous comment), in part because there is a sense amongst teachers (and this is a more anecdotal observation on my behalf) that non-digital manipulatives effectively do ‘just as good a job’ (and are less expensive, more accessible etc).
